# Fully Automatic Binary Glioma Grading based on Pre-Therapy MRI using 3D Convolutional Neural Networks

**Milan Decuyper**                                    milan.decuyper@ugent.be
and **Roel Van Holen**                                Roel.VanHolen@ugent.be
*Medical Image and Signal Processing (MEDISIP), Ghent University, Ghent, Belgium*

## Abstract

The optimal treatment strategy of newly diagnosed glioma is strongly influenced by tumour malignancy. Manual non-invasive grading based on MRI is not always accurate and biopsies to verify diagnosis negatively impact overall survival. In this paper, we propose a fully automatic 3D computer-aided diagnosis (CAD) system to non-invasively differentiate high-grade glioblastoma from lower-grade glioma. The approach consists of an automatic segmentation step to extract the tumour ROI followed by classification using a 3D convolutional neural network. Segmentation was performed using a 3D U-Net achieving a dice score of 88.53% which matches top performing algorithms in the BraTS 2018 challenge. The classification network was trained and evaluated on a large heterogeneous dataset of 549 patients reaching an accuracy of 91%. Additionally, the CAD system was evaluated on data from the Ghent University Hospital and achieved an accuracy of 92% which shows that the algorithm is robust to data from different centres.

**Keywords:** Deep Learning, CNN, Glioma Grading, MRI

## 1. Introduction

Tumour malignancy has a strong influence on therapy planning and prognosis of newly diagnosed glioma. Whereas a watch-and-wait policy can be opted in case of low-grade glioma, maximum safe resection combined with appropriate chemotherapy and radiotherapy is recommended by EANO for high-grade glioma (Weller et al., 2017). Differentiation between low- and high-grade glioma is usually based on MRI with gadolinium-based contrast agents. The presence of contrast enhancement and necrosis are indicative of higher tumour malignancy, however 40-45% of non-enhancing lesions are subsequently found to be highly malignant (Jansen et al., 2012). This results in reduced accuracy of non-invasive tumour grading (sensitivities ranging between 55% and 83%). Biopsies to confirm histopathological diagnosis negatively impact overall survival (Wijnenga et al., 2017) and hence accurate non-invasive grading is preferred. Most CAD methods for glioma grading reported today are not fully automatic and often trained and evaluated on a small dataset from one clinical centre (Yang et al., 2018).

In this study we designed a fully automatic computer-aided diagnosis system to non-invasively discriminate high-grade glioblastoma (GBM) from lower-grade glioma (WHO grade II and III) using convolutional neural networks that are trained and evaluated on a large multi-centre dataset.

## 2. Methods

### 2.1. Data

The data used in the work originates from both public datasets and data that was retrospectively acquired from the Ghent University Hospital, with permission of the local ethics committee (Belgian registration number B670201838395 2018/1500). The included public databases are the BraTS 2018 dataset (Menze et al., 2015; Bakas et al., 2017a) and the TCGA-GBM (Scarpace et al., 2016), TCGA-LGG (Pedano et al., 2016) and LGG-1p19qDeletion (Erickson et al., 2017) collections on The Cancer Imaging Archive (Clark et al., 2013). Inclusion criteria were: a histologically proven glioma of WHO grade II, III or IV and the availability of a preoperative T1ce MRI together with a FLAIR and/or T2 sequence of sufficient quality. In total 549 patients were acquired from the public databases and 112 patients from the University Hospital resulting in data from 660 patients. All MRI were co-registered, interpolated to 1 $mm^3$ voxel sizes, bias corrected and skull-stripped using SPM12. Each modality of every patient is independently normalised by subtracting the mean and dividing by the standard deviation.

### 2.2. Segmentation

The first part of the Grading system consists of segmenting the whole tumour volume. As U-Nets have shown state-of-the-art performance for brain tumour segmentation in recent BraTS challenges, we implemented a 3D U-Net similar to the architecture proposed in Isensee et al. (2018) with 25 features at the highest resolution, instance normalisation and leaky ReLUs. The network was trained on random patches of size 128x128x128, batch size of two, ADAM optimisation ($lr_{init} = 1 \cdot 10^{-4}$), soft dice loss and L2 weight decay of $10^{-5}$. From the 285 patients in the BraTS 2018 training data, 60 were used for validation. The network was finally evaluated on 76 patients from the TCGA-GBM and TCGA-LGG collections that were included in the BraTS 2018 test data (Bakas et al., 2017b). For patients from TCIA and the University Hospital, not all four MRI sequences (T1, T1ce, T2 and FLAIR) are always available. Therefore, during training, the T1 and the T2 or FLAIR channels are randomly set to zero to increase robustness to missing modalities.

### 2.3. Grading

After segmentation, the glioma is classified as a glioblastoma or a lower-grade glioma (WHO grade II or III). To this end, a tumour region of interest (ROI) is extracted from the T1ce MRI, resized to a size of 112x112x112 and subsequently fed into the classification network. The used network architecture (see Figure 1) consists of one convolutional layer with a kernel size of 7 followed by 4 residual blocks. Every convolutional layer is succeeded with instance normalisation and a ReLU layer. The network is trained using SGD ($lr_{init} = 1 \cdot 10^{-3}$, $momentum = 0.9$), L2 weight decay of $10^{-5}$ and a batch size of 8. To prevent overfitting, data augmentations like flipping, rotations, and padding were applied on the fly during training. Four hundred patients were used for training (200 GBM and 200 LGG cases), 69 for validation (35 GBM, 34 LGG) and 80 TCIA patients for final evaluation (34 GBM, 46 LGG). We made sure that test patients were not used in the training set of the segmentation network in order to evaluate the system on data that both parts have never seen before.

The University Hospital data (63GBM, 49 LGG) was used to test the performance on a final independent dataset. Both the segmentation and classification networks were implemented in PyTorch and trained on an 11GB NVIDIA GTX 1080 Ti GPU.

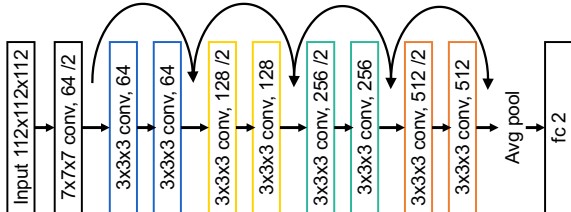

Figure 1: Architecture used to classify a tumour ROI as GBM or LGG. Every convolutional layer is succeeded with instance normalisation and a ReLU activation.

## 3. Results and Discussion

The obtained whole tumour dice scores on the 76 BraTS 2018 test patients were 88.53%, 86.38% and 84.62% when providing all four modalities, only T1ce and FLAIR and only T1ce and T2 sequences as input respectively. These scores match the performance of state-of-the-art algorithms in the most recent BraTS 2018 challenge (Myronenko, 2018). Small variations between manual and predicted segmentations won't have a strong influence on the tumour ROI, making the obtained performance sufficient for the current task.

The area under the ROC curve (AUC), matthews correlation coefficient (MCC), accuracy, sensitivity (percentage of GBM cases that are correctly classified as such) and specificity scores on the TCIA test set and University Hospital data are shown in Table 1. These results indicate that, to the best of the authors' knowledge, state-of-the-art grading performance is achieved using a system that is trained on a large heterogeneous dataset and is fully automatic. Moreover, the strong performance on independent data from the Ghent University Hospital shows that it is robust to variations in imaging protocols and data from different clinical centres.

Table 1: Results of the binary glioma grading system on both data from TCIA and the Ghent University Hospital

| Dataset | AUC | MCC | Acc. | Sens. | Spec. |
|---|---|---|---|---|---|
| TCIA Test Data | 96.29 | 82.19 | 91.25 | 91.18 | 91.30 |
| Ghent Univeristy Hospital Data | 93.39 | 83.91 | 91.96 | 90.48 | 93.88 |

## 4. Conclusion

In this paper we presented a fully automatic 3D approach to classify glioma as a high grade glioblastoma or lower-grade glioma based on pre-therapy structural MRI which has important value for optimal therapy planning and prognosis. The two-step procedure consisting of segmentation using a 3D U-Net and classification with a 3D residual network achieved state-of-the-art results on a large heterogeneous dataset and generalises well to data from different centres.

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
