# OpenReview forum: "Fully Automatic Binary Glioma Grading based on Pre-Therapy MRI using 3D Convolutional Neural Networks"
_MIDL.io/2019/Conference/Abstract — MIDL Abstract 2019_

### Official Review · AnonReviewer2 · 2019-04-29
**The benefit of the proposed approach is unclear**

**Rating:** 2
**Confidence:** 2

**Review:**

The paper shows that the proposed two step approach generalizes well for data with different sites. While the result is satisfactory, it is not the first time that such multi-site method is presented, and since one needs to combine the dataset anyways, I don't see the benefit over just training on each dataset. In the given presentation, it is also not clear if multi-site method has significant performance difference compared to the ones trained on homogeneous dataset.

---

### Official Review · AnonReviewer1 · 2019-05-01
**Interesting clinical application using best state-of-the-art practices to develop an effective fully automated pipeline for glioma grading.**

**Rating:** 3
**Confidence:** 2

**Review:**

A 3D fully automated computer-aided diagnosis scheme for the differentiation of high-grade glioblastoma and lower-grade glioma is presented.

The proposed pipeline is rather straightforward. In a first step, a 3D segmentation network is employed to segment whole tumor regions in the image. The authors report high segmentation performance at this stage which allows extraction of the tumor regions, for the next step. At this second step, a 3D classification network for glioma grading is used to classify the extracted tumors.

The method has been evaluated on several datasets and shows convincing results.

Overall, this is an interesting clinical application. Using the best state-of-the-art practices, the authors were able to develop an effective fully automated pipeline for glioma grading.

---

### Decision · Program_Chairs · 2019-05-06
**Acceptance Decision**

Accept